# Isolation and Functional Characterization of *MsFTa*, a *FLOWERING LOCUS T* Homolog from Alfalfa (*Medicago sativa*)

**DOI:** 10.3390/ijms20081968

**Published:** 2019-04-22

**Authors:** Junmei Kang, Tiejun Zhang, Tao Guo, Wang Ding, Ruicai Long, Qingchuan Yang, Zhen Wang

**Affiliations:** Institute of Animal Science, Chinese Academy of Agricultural Sciences, Beijing 100193, China; kangjunmei@caas.cn (J.K.); tiejunzhang@126.com (T.Z.); Yushen0002008@126.com (T.G.); dingwang@126.com (W.D.); dragongodsgod@163.com (R.L.); qchyang66@163.com (Q.Y.)

**Keywords:** alfalfa, *MsFTa*, flowering time, transgenic plants

## Abstract

The production of hay and seeds of alfalfa, an important legume forage for the diary industry worldwide, is highly related to flowering time, which has been widely reported to be integrated by *FLOWERING LOCUS T* (*FT*). However, the function of *FT*(s) in alfalfa is largely unknown. Here, we identified *MsFTa*, an *FT* ortholog in alfalfa, and characterized its role in flowering regulation. *MsFTa* shares the conserved exon/intron structure of *FTs*, and the deduced MsFTa is 98% identical to MtFTa1 in *Medicago trucatula*. *MsFTa* was diurnally regulated with a peak before the dark period, and was preferentially expressed in leaves and floral buds. Transient expression of MsFTa-GFP fusion protein demonstrated its localization in the nucleus and cytoplasm. When ectopically expressed, *MsFTa* rescued the late-flowering phenotype of *ft* mutants from Arabidopsis and *M. trucatula*. *MsFTa* over-expression plants of both Arabidopsis and *M. truncatula* flowered significantly earlier than the non-transgenic controls under long day conditions, indicating that *exogenous MsFTa* strongly accelerated flowering. Hence, *MsFTa* functions positively in flowering promotion, suggesting that *MsFTa* may encode a florigen that acts as a key regulator in the flowering pathway. This study provides an effective candidate gene for optimizing alfalfa flowering time by genetically manipulating the expression of *MsFTa*.

## 1. Introduction

Flowering time, the switch from vegetative to reproductive growth, is a key trait for the seasonal and geographical adaptation of flowering plants. The mechanism of flowering is best understood in the annual plant Arabidopsis, in which the timely onset of flowering is regulated by an intricate network with various genetic regulators responding to endogenous (gibberellin, autonomous and aging) or environmental (vernalization and photoperiod) triggers, as reviewed in recent years [1,2,3,4]. The balance of signals from these pathways is integrated by a common set of genes, including *FLOWERING LOCUS C (FLC)/MADS AFFECTING FLOWEIRNG (MAF), FLOWERING LOUCS T* (*FT*), *CONSTANS* (*CO*) and *SUPPRESSOR OF OVEREXPRESSION OF CONSTANS 1* (*SOC1*), to determine the flowering time [5,6,7]. In *Medicago*, however, *FLC/MAF* clade flowering repressors are missing and *CO*-like genes do not seem to regulate flowering [8,9,10]. Hence, florigen encoding gene *FT*, acting as a long-distance hormone signal between leaves and shoot apical meristems, is the most important flowering integrator of the legume species.

Since the search for the flowering hormone florigen in the 1930s, an increasing number of *FT*-like genes have been identified in a wide range of annual plants, such as rice, soybean and barrel clover [11,12,13]. For example, in *M. truncatula* five *FT*-like genes, namely *MtFTa1*, *MtFTa2*, *MtFTb1*, *MtFTb2* and *MtFTc*, have been characterized. Among them, *MtFTa1*, *MtFTb1* and *MtFTc* rescued the late flowering phenotype of Arabidopsis *ft-1* mutant, while the other two genes failed to act as a florigen [11]. Functional differentiation among *FT* paralogs derived from lineage-specific duplication has been observed in several species, such as soybean, potato and rice [13,14,15]. The diversity of the functions of *FT* paralogs is attributed to the expression pattern shift and the divergence in gene regulatory networks.

In recent years, studies on *FT*-integrated flowering regulation have focused on economically important perennial plants including grasses (e.g., switchgrass and ryegrass) [16,17] and trees (e.g., poplar, apple and cherry) [18,19,20]. In contrast to annual plants, *FT* is expressed in both leaves and flowers of several perennial species such as sweet cherry, kiwifruit and leafy spurge [21,22,23]. Functional coordination of perennial *FT* paralogs revealed the evolution of a complex perennial adaptive trait after genome duplication. For example, poplar *FT1* determined the onset of reproductive stage in winter, whereas *FT2* promoted vegetative growth and inhibited bud set in the growing season [18]. Ectopic over-expression of *FT* perennial orthologs in Arabidopsis caused early flowering in most cases, suggesting a conserved function in perennial species [16,17,18,19,20,22].

Alfalfa (*M. sativa*), one of the most important forages globally, exhibits seasonal flowering. Different from perennial woody plants, which are not able to form flower buds during the first several years of their life cycle, herbaceous alfalfa grows vegetatively at early developmental stages and then switches to the reproductive stage in the same year. In Northern China, a main alfalfa planting region in the country, the first cutting coincides with the rain season, which severely limits hay production. Breeding alfalfa cultivars with optimal flowering time would facilitate the improvement of the quantity and quality of alfalfa. Our study here demonstrated that *MsFTa*, the first isolated *FT* from alfalfa, not only shared the conserved functional domains of its orthologs, but also acted ectopically in accelerating flowering. These findings would supply a potential candidate for the generation of alfalfa cultivars with an optimal flowering time for the main planting regions in Northern China via genetic modulation of *MsFTa* expression.

## 2. Results

### 2.1. MsFTa Sequence Shared the Common Features of FT-like Genes

*FLOWERING LOCUS T* (*FT*) genes have been intensively documented in many plants including *M. truncatula*, a model annual legume species [8,9,10,11]. Up till now, five *MtFTs*, namely *MtFTa1*, *MtFTa2*, *MtFTb1*, *MtFTb2* & *MtFTc*, have been identified from *M. truncatula*. Using *MtFTa1* as a query sequence, we did a BLAST search against the most updated *M. truncatula* genome database (http://plants.ensembl.org/Medicago_truncatula) and found that besides the four *MtFT* paralogs, one more gene (*Mt6g033040*) annotated as *Flowering Locus Protein T* (*MtFT*) was hit with a homology of 64–73% to its paralogous proteins (Appendix A). Thus, the *M. truncatula* genome encodes at least six *FT*-like genes with sequence similarity higher than 60% (Figure 1).

To investigate flowering regulation in perennial legume alfalfa, we cloned an *FT*-like cDNA from the forage by combination of RT-PCR and RACE strategy based on the sequence information of *MtFTs*. The 1029 bp cDNA sequence contained an open reading frame (ORF) of 531 bp (GenBank: JF681135) (Appendix A). The putative protein consists of 176 amino acids with theoretical isoelectric point (pI) of 8.23 and molecular weight of 19.7 kDa, respectively. Phylogenetic analysis of 33 FT and FT-like proteins from rice, barley, Arabidopsis, soybean, barrel clover and poplar demonstrated that the six MtFTs were grouped into three subclasses, namely Subclass A (MtFTa1, MtFTa2 & Mt6g033040), Subclass B (MtFTb1 & MtFTb2) and Subclass C (MtFTc) (Figure 1). The putative alfalfa FT protein was clustered into Subclass A sharing the same branch with MtFTa1 and MtFTa2 (Figure 1). Hence, it was designated as MsFTa, the first *FLOWERING LOCUS T* gene identified in alfalfa. Compared with Subclass A members from Arabidopsis, rice and barley, MsFTa was more closely related to the FTs of legume plants such as barrel clover and soybean (Figure 1). Sequence homology analysis showed that MsFTa shared an identity of 59–99% with MtFT members, the lowest identity with MtFTc and the highest with MtFTa1, respectively (Appendix A). The sequence similarity of MsFTa with FTs from the other five species ranges from 54% to 84% (Appendix A). Multiple alignment revealed that, like its orthologs, MsFTa has a ligand-binding motif with a conserved key residue Tyr (Y) at position 85, an external loop of a 14-amino acid segment (residues 128–141) and a triad (residues 150–152) at the C-terminal end (Figure 2) [24,25,26]. In addition, MsFTa features a large number of alpha helices, beta-folds and random coil structures (Appendix A). Our prediction using ScanProsite showed that most regions of MsFTa are hydrophilic (Appendix A), indicating it is a hydrophilic protein.

Comparison of the exon–intron structure of *FT*-like genes demonstrated that like most *FT*-like genes from a variety of species [11,13,27,28], *MsFTa* is composed of four exons separated by three introns (Figure 3). Interestingly, exon 2 is universally 62 bp and exon 3 is 41 bp, except two barley *FT*-like genes (*Hv3Hr1g0871002* and *Hv7Hr1g0246102*) (Figure 3). The 2nd and 3rd exons of *MsFTa* encode the N-terminal of the ligand-binding motif, including the amino acids forming the ligand-binding pocket and the key residue Tyrosine (Y) at position 85 (Figure 2). Therefore, *MsFTa* not only has the same exon composition with conserved open reading frame but also possesses the FT functional domains shared by a wide range of plant species.

### 2.2. The *MsFTa-GFP* Recombinant Protein Resided in Both the Cytoplasm and Nucleus

To examine the subcellular localization of *MsFTa*, *35S::MsFTa-GFP* and *35S::GFP* were transformed into onion epidermal cells separately. The transient expression of the GFP control was uniformly distributed throughout the cell, whereas the MsFTa-GFP fusion protein was observed in the nucleus and cytoplasm, especially the membrane (Figure 4). This result is consistent with the recent observations of FT-like proteins in perennial species such as switchgrass and sweet cherry [16,22].

### 2.3. Diurnal and Tissue-Specific Expression Pattern of MsFTa

To monitor the temporal and spatial expression profiles of *MsFTa*, quantitative RT-PCR was performed using *MsFTa*-specific primers. Analysis of the *MsFTa* transcript demonstrated that the level of *MsFTa* progressively increased after 6 h light and reached a peak at 16 h light. The onset of darkness caused a sharp deduction of *MsFTa* and the descending trend continued during the night period (Figure 5a). Spatially, four tissues from both underground and aerial organs were tested. *MsFTa* was detected in all tested tissues including roots, stems, floral buds and leaves with the highest expression in leaves, which is approximately 3.4-fold of the amount in roots (Figure 5b). The expression pattern of *MsFTa* is highly reminiscent of its legume homologs, especially *MtFTa2* in *M*. *truncatula* [11]. Detailed analysis demonstrated that besides cauline leaves, a higher level of *FT* transcript was detected in flowers, siliques, and seeds at early stages (http://bbc.botany.utoronto.ca/efp) (Appendix A).

### 2.4. Ectopic Expression of MsFTa in Arabidopsis Strongly Accelerated Flowering

To explore the biological function of *MsFTa* in modulating flowering, the construct of *MsFTa* driven by the cauliflower mosaic virus 35S promoter was introduced separately into wild type Arabidopsis and *ft-10*, a T-DNA insertion mutant of *FT*, by agrobacterium infiltration. The existence of a *35S::MsFTa* construct in two independent transgenic *Arabidopsis* T2 lines (T2-1 & T2-4) with kanamycin resistance was confirmed by PCR (Figure 6a). Semi-quantitative RT-PCR of the over-expression lines demonstrated that *MsFTa* transcript was abundantly expressed in both transgenic lines while absent in wild type Arabidopsis (Figure 6b). Phenotypically, the over-expression plants exhibited early flowering (Figure 6c). The measurement of leaf number showed that wild type plants flowered with 15.4 leaves, whereas T2-1 and T2-4 over-expression lines flowered with 12.1 and 11.4 leaves, respectively (*p* < 0.05) (Figure 6d). Consistently, in terms of days after germination (DAG), the first flower of line T2-1 and T2-4 appeared at 16.1 and 17.8 DAG, respectively, while the first flower of the wild type was observed at 24.3 DAG (*p* < 0.05) (Figure 6e). These results suggested that flowering time of the transgenic Arabidopsis was associated with *MsFTa* transcript level.

Arabidopsis mutant *ft-10* is a strong allele showing late-flowering phenotype. Under LD, *ft-10* flowered when the mutant plant had 36.8 leaves, while wild type flowered with 15.4 leaves (*p* < 0.05) (Figure 6f,g). For the transgenic plants in *ft-10* background, eight out of 10 independent T1 lines flowered at a time similar to that of wild type. Flowering time analysis of a representative T2 line showed that the *MsFTa* transgenic *ft-10* flowered with 16.5 leaves (Figure 6f,g), indicating that introduction of *35S::MsFTa* complemented the late-flowering phenotype of *ft-10* mutant. Except flowering time, the transgenic plants did not display obvious morphological difference. Taken together, our findings demonstrated that constitutive expression of alfalfa *MsFTa* in Arabidopsis resulted in early flowering, suggesting a positive role of *MsFTa* in flowering regulation.

### 2.5. Ectopic Expression of MsFTa Promoted Flowering in Transgenic M. truncatula Plants

Given the close phylogenetic relationship and high sequence identity of *MsFTa* to *MtFTa1*, both barrel clover wild type R108 and *mtfta1-1*, a late-flowering mutant (NF3307) caused by the insertion of a *Tnt1* retrotransposon in the first exon of *MtFTa1* [11], were transformed with the construct of *35S::MsFTa* mentioned above. Flowering time of the transgenic plants was measured as days to produce the first flower under LD conditions. Compared with R108 which flowered at D56, the two independent T1 lines (T1-1 & T1-2) overexpressing *35S::MsFTa* produced the first flower at D35 and D32, respectively. Over-expression of *MsFTa* in *M. truncatula* resulted in statistically significant early flowering (*p* < 0.05) (Figure 7a). For *mtft1-1* plants expressing *35S::MsFTa*, the flowering time of two independent transgenic lines was analyzed. As shown in Figure 7b, the first flower of the two lines emerged at D58 and D61, while *fta1-1* flowered at D83, which is significant later than R108 at D56 (*p* < 0.05), indicating that *MsFTa* successfully rescued the late flowering phenotype of *fta1-1*. These results suggested that similar to ectopic expression of *MsFTa* in Arabidopsis, introduction of *MsFTa* into *M. truncatula* caused strong flowering promotion.

## 3. Discussion

Precise flowering time ensures that the onset of flowering occurs at an ideal time to optimize plant biomass and fitness in specific environmental conditions. Although diverse upstream components of flowering pathways have been identified from plants, *FLOWERING LOCUS T* appears to be a universal regulator ultimately integrating flowering signaling in crops, vegetables, grass and trees [11,18,26,30]. In this study, we provided molecular and genetic evidence that *MsFTa*, an *FT* homolog from alfalfa, functioned to accelerate flowering.

As a member of an ancient phosphatidylethanolamine-binding protein (PEBP) gene family, *FT* has a conserved exon/intron structure and highly identical protein sequence across species [13,31]. Indeed, our results demonstrated that *MsFTa* possessed the characteristic features of a classic *FT* gene. First, *MsFTa*, similar to *FT*-like genes, has four exons with conserved exon/intron boundaries [13,27,28]. Over-expression of the chimeric genes constructed by replacing individual exons of *FT* revealed that substitution of the fourth exon, especially the external loop caused late flowering, presumably the crystal structure was affected due to the sequence variation of the exon [24]. It has been reported recently that exons two and three were also crucial for the functional conversion of FT in Arabidopsis [13,32]. Second, MsFTa shared several well characterized amino acid residues proven to be important for flowering acceleration according to the experiments of individual residue exchange or segment replacement. For example, in Arabidopsis, alteration of Tyr 85 (Y85H) changed FT functions in promoting flowering, indicating that the absolutely conserved Tyr 85 is a key residue determining the protein function as an activator of flowering [25]. *FTs* share a triad ending with residue asparagine (N) at site 152, followed by leucine (Figure 2). The crystal structure revealed that the triplet motif contacts the adjacent external loop, which facilitates access to FT interactors, such as the bZIP transcription factor FD and 14-3-3 proteins [24,33,34,35]. In Rosaceae species, mutations of the triad prevented the *FT*-like genes from promoting flowering [36]. Third, *FTs* have highly conserved motifs including a potential ligand-binding pocket and an external loop (Figure 2), both of which were proven to contribute to the functional specificity of FT [24,33]. Compared with other species, it seems that the external loop of legume plants is variant (Figure 2), implying varied conformation, possibly leading to flexible accessibility to a binding partner. Therefore, *MsFTa*, together with *FT*-like genes, specifies FT conformation by possessing identical exon/intron structure and highly conserved functional motifs.

*FT*-like orthologs from a variety of plant species have been shown to promote flowering when ectopically expressed [16,22,37,38,39]. In the model legume *M. truncatula*, reverse genetics revealed that the loss-of-function mutant of *MtFTa1*, the closest homolog of *MsFTa*, caused late flowering and *MtFTa1* fully complemented *fta1-1* defects [11]. Consistently, an allelic series of gain-of-function mutations in *MtFTa1* caused by retroelement insertions exhibited a dominant early flowering phenotype [40,41], confirming its function in flowering regulation. It appears that similar to *FT*, *MsFTa* from the perennial forage acts positively in flowering promotion. The notion is supported directly by the transgenic Arabidopsis and *M. truncatula* including both complementation and over-expression plants. Plants ectopically expressing *MsFTa* in either wild type or *ft* mutant of both plant species displayed a clear shift to early flowering (Figure 6d–g). Statistically, the flowering time difference between the transgenic plants and the non-transgenic ones is significant (*p* < 0.05). Notably, compared with wild type, complementation plants flowered slightly late, suggesting that constitutive expression of *MsFTa* partially rescued the flowering time of *ft* mutants from Arabidopsis and barrel clover. These findings imply a minute functional distinction between *MsFTa* and its orthologs, *FT* and *MtFTa1*, in Arabidopsis and barrel clover, respectively. Application of the endogenous promoter of *MsFTa* to produce transgenic alfalfa is undergoing. Moreover, generation of a loss of function mutant of *MsFTa* using the CRISPR-Cas9 strategy would be necessary to elucidate the endogenous function of *MsFTa* in alfalfa flowering regulation.

Given the detection of *MsFTa* in roots and stems, it is tempting to speculate that besides flowering promotion, *MsFTa* may participate in other development processes of alfalfa. *FT* orthologs have been documented to be involved in multiple functions like plant architecture [42], tuberization [15], and dormancy interruption [43]. The expression of *MsFTa* resembled *FT* orthologs in perennial plants, such as sweet cherry, loquat, leafy spurge and tea-oil tree, in which *FT*-like genes are expressed in diverse tissues, including mature leaves, floral buds and reproductive organs (flower organs and fruits) [21,22,27,44]. The expression divergence may be an important force in driving the functional division of *FTs* in perennial plants, especially in species with multiple *FT* paralogs [13,14,16,22]. As it is closely related to diploid legume *M. truncatula*, which has six identified *FT*-like genes, tetraploid alfalfa is more likely to have a similar number, or even more, *MsFTa* paralogs due to genome duplication. Long-term work of flowering regulation in alfalfa will be directed at paralogs of *MsFTa*.

## 4. Materials and Methods

### 4.1. Plant Material and Growth Conditions

*Medicago sativa* cv. ‘Zhongmu No.1’, bred by the Institute of Animal Science, the Chinese Academy of Agricultural Sciences, was used in this study. Seeds were germinated in regular soil in a 20 cm diameter pot. *Medicago truncatula Tnt1* insertion mutant *fta1-1* (NF3307) was obtained from the Noble Research Institute (Ardomor, OK, USA). *Arabidopsis thaliana* seeds of *ft-10* (CS9869) were from Arabidopsis Biological Research Center (ABRC) and germinated in soil (vermiculite:soil = 1:3). All plants were grown in growth chambers at 21 °C under a 16/8 h (light/dark) photoperiod.

### 4.2. Cloning of MsFTa from Alfalfa

Total RNA was extracted from 2-week-old alfalfa seedlings with TRIzol reagent (Sangon, Shanghai, China). After DNase I (Fermentas-MBI, Shenzhen, China) treatment, the first-strand cDNA was synthesized using Reverse Transcriptase III (Takara, Dalian, China). A DNA fragment was amplified by using the two degenerate primers designed based on *FT*-like genes in *M. truncatula* and sequenced. The obtained partial sequence of alfalfa *FT*-like gene was used to design primes for amplication the full-length *MsFTa* by 3′-RACE and 5′-RACE (Invitrogen, Carlsbad, CA. USA) with the specific primers according to the manufacturer’s instructions. PCR amplicons were purified after separated on 1% agarose gel and cloned into pMD19-T (TaKaRa, Dalian, China) and sequenced.

### 4.3. Bioinformatic Analysis

For *FT* homologous genes, Arabidopsis FT (*At1g65480*) was used as query sequence to search against the database EnsemblPlants (http://plants.ensembl.org/index.html). Phylogenetic analysis was performed with bootstrap trials of 1000 using DNAMAN version 7.0 (Lynnon Corporation, Quebec, QC, Canada), and sequence alignment was analyzed using the same software with default parameters. Isoelectric point (pI) and molecular weight (MW) values were predicted using ExPaSy (https://prosite.expasy.org). The secondary structure of the protein was predicted using LYON-GERLAND PBIL (https://prabi.ibcp.fr); and the hydrophobic structure was predicted using ScanProsite (https://prosite.expasy.org/scanprosite).

### 4.4. Expression Analysis by RT-PCR

For expression analysis in different tissues, roots, stems and leaves were harvested from 6-week-old alfalfa, while flowers were from 60-day-old plants. For the diurnal analysis, leaves were collected separately from 6-week-old alfalfa at zeitgeber time (ZT) 0, 2, 4, 6, 8, 10, 12, 14, 16, 18, 20, 22 and 24 time points, respectively. After extraction of total RNA, cDNA was prepared as described above. Semi-quantitative RT-PCR was carried out using gene-specific primers with *MsActin* serving as an internal control. The PCR product was analyzed in gel and images were captured using a Gel Image Analysis System FR-200A (Furi Science & Technology, Shanghai, China). For quantitative real-time PCR, SYBR Premix EX Taq^TM^ qPCR Kit (Takara, Dalian, China) was used according to the instruction with BIO-RAD CFX96^TM^ Real-Time System (BioRad, Hercules, CA, USA). The Arabidopsis *Actin* gene was used as a loading control. The sequence of the primers used in this study is listed in Appendix A.

### 4.5. Subcellular Localization Analysis

For the subcellular localization analysis of MsFTa-GFP fusion protein in onion epidermal cells, the construct of MsFTa-GFP was performed as follows: the *MsFTa* ORF sequence was subcloned into the pA7-GFP (35S::GFP) expression vector [29] using the primers of FT-GF and FT-GR containing *XhoI* and *SpeI* restriction sites, respectively. The plasmid of the pA7-GFP and 35S::MsFTa-GFP fusion binary vector was then transformed into separate onion epidermal cells via particle bombardment (GJ-1000, Scientz Biotechnology, Ningbo, China). The subcellular localization of the transiently expressed MsFTa-GFP fusion protein was imaged using a confocal laser-scanning microscope (Olympus FV500, Tokyo, Japan).

### 4.6. Plasmid construct and plant transformation

For generation of the transgenic Arabidopsis plants expressing *35S::MsFTa*, first, a binary vector harboring *35S::MsFTa* was constructed by amplifying *MsFTa* ORF with primers named pBI-FTa-f and pBI-FTa-r, and the sequence-confirmed *MsFTa* was introduced into the pBI121 vector (Clontech, Palo Alto, CA, USA) using *Xba* I and *Sma* I restriction sites. Secondly, the *35S::MsFTa* plasmid was transformed into *Agrobacterium tumefaciens* strain GV3101 by electroporation, and *Arabidopsis* transformation was performed via the floral-dip method. T1 *Arabidopsis* seeds were screened on a MS medium containing 40 mg·L^−1^ kanamycin, and T2 lines confirmed by semi-quantitative RT-PCR were used in this study for *MsFTa* expression and phenotypic analysis. For Medicago transformation, explants from 4-week-old plants (R108 and *fta1-1*) growing under LDs were cocultivated with the agrobacterium containing the construct of *35S::MsFTa* according to the Medicago handbook [45]. Kanamycin of 40 mg·L^−1^ was used to select transformant plants.

## 5. Conclusions

In this study, the *FT* homolog from alfalfa, one of the most important perennial forages, was isolated and characterized. *MsFTa* has highly identical sequence composition, especially exon/intron boundary and functional domains, with *FTs* in a variety of plant species. The diurnally regulated *MsFTa* is expressed preferentially in leaves and floral buds. Functionally, *MsFTa* accelerated flowering in Arabidopsis and the annual legume *M. truncatula*, indicating that the novel alfalfa *FT* acts as a flowering promoter.

## Figures and Tables

**Figure 1 ijms-20-01968-f001:**
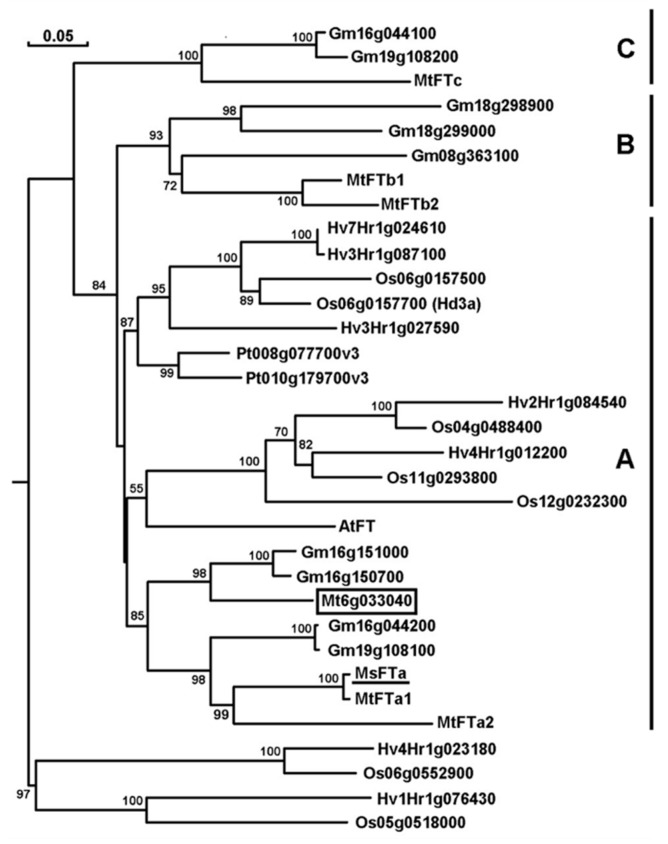
Phylogenetic analysis of MsFTa and its orthologs in the indicated plant species. The full-length sequence of 33 FT and FT-like proteins from six model species was analyzed using DNAMAN (version 7.0). The numbers at the nodes of the neighbor-joining tree indicate the bootstrapping values based on 1000 interactions. *MsFTa* was underlined and Mt6g033440, a novel FT in *M. truncatula* was boxed. Gene accession numbers: *MsFTa* (GenBank: JF681135); *Arabidopsis thaliana* (At): *AtFT* (*At1g65480*); *Medicago truncatula* (Mt): *MtFTa1* (*Mt7g084970*), *MtFTa2* (*Mt7g085020*), *MtFTb1* (*Mt7g006630*), *MtFTb2* (*Mt7g006690*), *MtFTc* (*Mt7g085040*), *MtFT* (*Mt6g033040*); *Glycine max* (Gm): *Gm19g108100*, *Gm18g298900*, *Gm18g299000*, *Gm16g151000*, *Gm16g044200*, *Gm16g150700*, *Gm16g044100*, *Gm19g108200*, *Gm08g363100*; *Populus trichocarpa* (Pt): *Pt8g077700*, *Pt10g179700*; *Hordeum vulgare* (Hv): *Hv1Hr1g076430*, *Hv2Hr1g023180, Hv2Hr1g084540*, *Hv3Hr1g027590*, *Hv3Hr1g087100*, *Hv4Hr1g012200*, *Hv7Hr1g024610* and *Oryza sativa* (Os): *OsHd3a* (*Os06g0157700*), *Os04g0488400*, *Os05g051800*, *Os06g0157500*, *Os06g0552900*, *Os11g0293800*, *Os12g0232300*.

**Figure 2 ijms-20-01968-f002:**
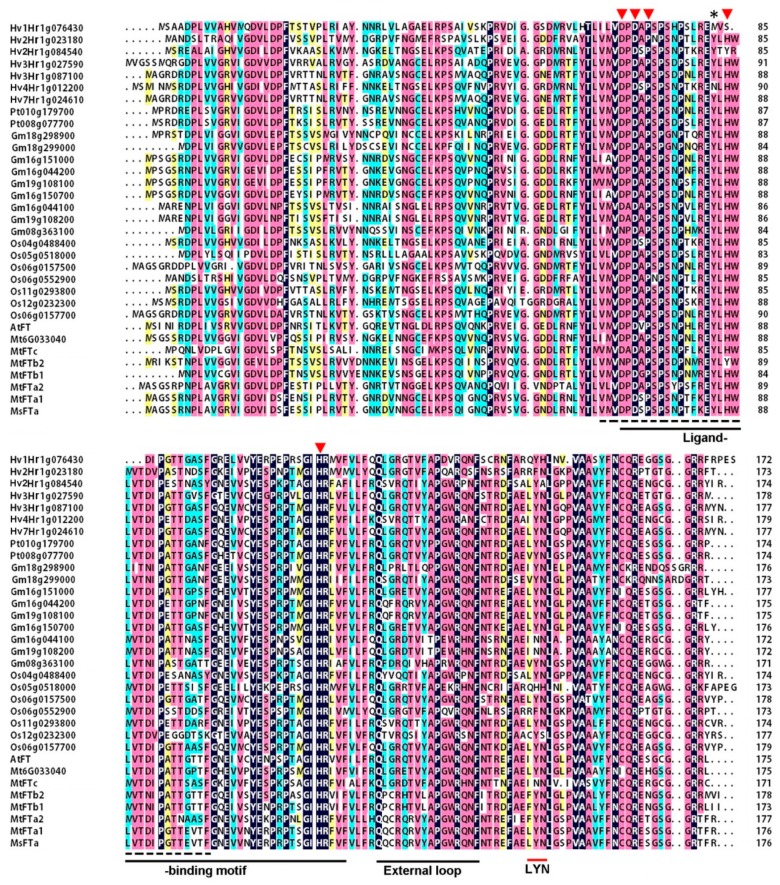
Sequence alignment of FTs in the indicated species. Sequence was aligned using DNAMAN Version 7 (Lynnon Corporation, Quebec, Canada). Homology level was highlighted by shading in color: black for 100%, pink for ≥75%, blue for ≥50% and yellow for ≥33% identity. The ligand-binding motif and the external loop of 14-amino acid stretch were underlined. The amino acid residues encoded by exon 2 (62 bp) and exon 3 (41 bp) were underlined with a dashed line. Triangles indicated the conserved amino acids in the ligand-binding pocket. An asterisk (*) indicated the conserved key residue Tyrosine (Y) at position 85 of *MsFTa*.

**Figure 3 ijms-20-01968-f003:**
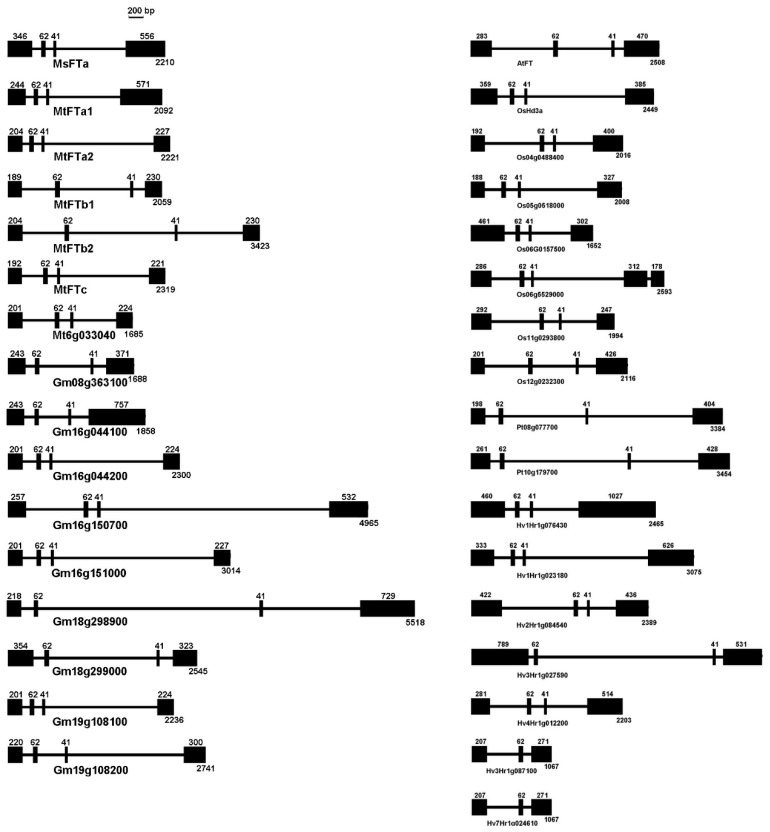
Schematic representation of the exon/intron structure of *FTs* in the indicated species. The information of *FT* gene composition is from EnsemblPlants (http://plants.ensembl.org). Gene structure was drawn using DNAMAN (Version 7). Solid lines represent introns and black boxes represent the exons with length indicated.

**Figure 4 ijms-20-01968-f004:**
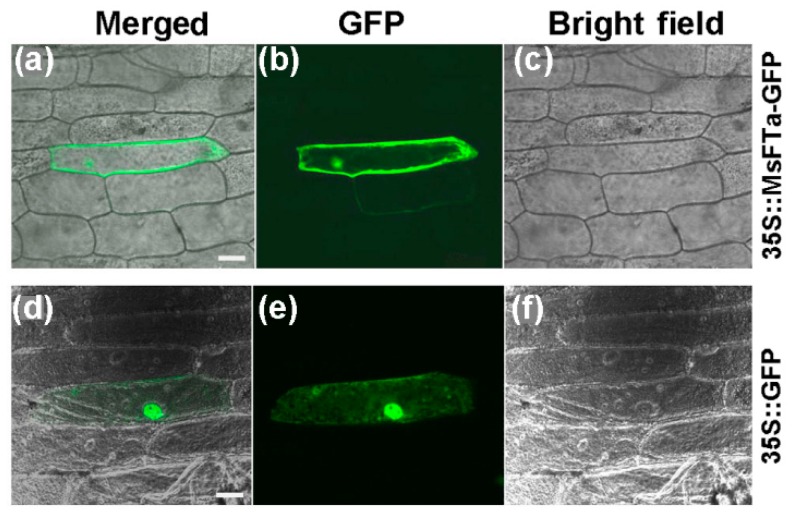
Subcellular localization of the MsFTa-GFP fusion protein in onion epidermal cells. The *MsFTa-GFP* fused construct and control vector pA7-GFP [29] were transiently transformed into onion epidermal cells by microprojectile bombardment. All images were captured using a confocal laser scanning microscope (Olympus FV500). (**a**–**c**): Image of onion epidermal cells expressing 35S::MsFTa-GFP taken under GFP fluorescence (**b**) or in bright field (**c**), the merged image was shown in (**a**); (**d**–**f**): Image of onion epidermal cells expressing 35S::GFP taken under GFP fluorescence (**e**) or in bright field (**f**), the merged one was shown in (**d**). Bar = 100 µm.

**Figure 5 ijms-20-01968-f005:**
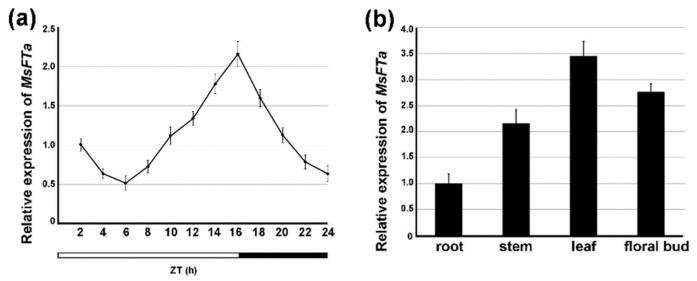
Diurnal and tissue-specific expression of *MsFTa.* (**a**) The expression level of *MsFTa* transcript during 24 h zeitgeber time (ZT) under LD. The white bar represents day time and the black bar represents night. (**b**) Relative expression of *MsFTa* in roots, stems, leaves and flowers of alfalfa at ZT 14. Bars represent the means ± SD of three biological replicates.

**Figure 6 ijms-20-01968-f006:**
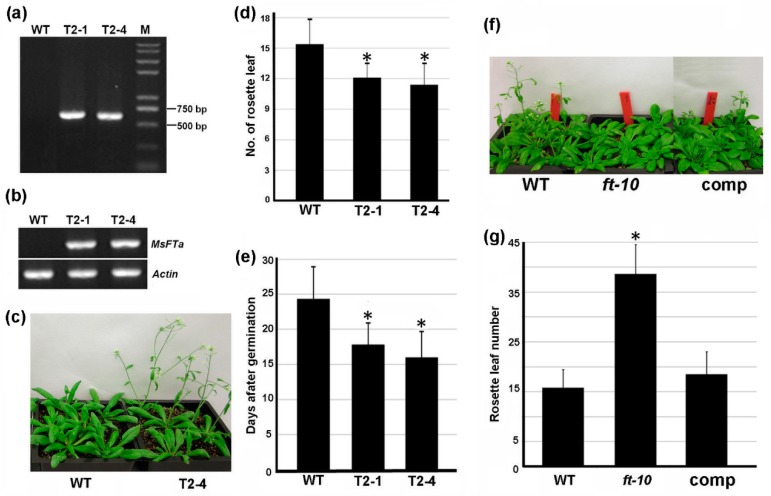
Ectopic expression of *MsFTa* in Arabidopsis caused early flowering. (**a**) Verification of two independent transgenic Arabidopsis lines (T2-1 and T2-4 of the T2 generation) harboring *35S::MsFTa* construct by PCR. (**b**) Transcriptional analysis of *MsFTa* in the two transgenic lines verified in (**a**). Arabidopsis *Actin 2* (*AT3g18780*) was used as internal control. Leaves from two-week old plants were harvested for total RNA extraction. (**c**) Three-week-old plants of Col-0 and *MsFTa* over-expressing line (T2-4). (**d**,**e**) Flowering time analysis of the over-expression T2 plants grown under LD conditions in terms of days after germination (**d**) and the No. of rosette leaves (**e**). At least 15 plants from three batches were scored for each line. The significant difference between the two genotypes was indicated as * (Student’s *t*-test, *p* < 0.05). (**f**) Four-week-old plants of Col-0, *ft-10* and the transgenic *ft-10* expressing *35S::MsFTa* (comp), showing rescued flowering time. (**g**) Statistical analysis of the No. of rosette leaves upon the emergence of the first flower. * indicated the significant difference from wild type (Student’s *t*-test, *p* < 0.05).

**Figure 7 ijms-20-01968-f007:**
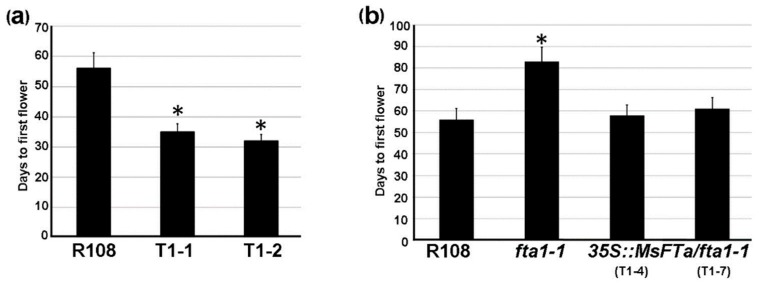
Ectopic expression of *MsFTa* in *M. truncatula* caused early flowering. (**a**) Comparison of flowering time between Medicago cv R108 overexpressing *35S::MsFTa* (T1) and control (R108) plants under LD conditions. (**b**) Flowering time analysis of *M. truncatula fta1-1* mutant (NF3307) plants ectopically expressing *35S::MsFTa* (T1) under LD conditions. Flowering time was measured as days to the emergence of the first flower under LD conditions. Two independent lines for each transformation were analyzed. Data represent average ± SD of three biological replicates. * indicates a significant difference from wild type (Student’s *t*-test, *p* < 0.05).

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
