# Peer review of "Isolation and Functional Characterization of MsFTa, a FLOWERING LOCUS T Homolog from Alfalfa (Medicago sativa)"

_ijms, 2019, doi:10.3390/ijms20081968_

Round 1

Reviewer 1 Report

I had an opportunity to review manuscript “Isolation and Functional Characterization of MsFTa, a FLOWERING LOCUS T homolog from Perennial Forage Alfalfa (Medicago sativa)” for future publication in International Journal Molecular Science. This manuscript by Kang and coauthors describes the key role of FT ortholog in alfalfa in flowering time. Authors performed a functional characterization of MsFTa by expression analysis, subcellular localization studies and generation of over-expression lines.

The project is correctly addressed, results obtained imply interesting conclusions and the manuscript is clear, easy to read and concrete. But minor revisions should be addressed.

-Line 83_94: This paragraph must be included in the legend of figure 1.

-Line 118_123: This paragraph must be included in the legend of figure 2.

-Line 134_136: This paragraph must be included in the legend of figure 3.

-Line 150_153: This paragraph must be included in the legend of figure 4.

-Line 148_150: These lines must be included in the material and methods section.

-Line 172_173, 192_193, 194_196:  Materials and Methods are described in these lines, which are duplicated in the material and methods section.

-Line 141_144: These sentences must be included in discussion section.

-Line 162_167: These sentences must be included in discussion section.

-Line 215: Mutant fta1-1 should be briefly described in order to make easier the reading of results.

-Line 281: “…a loss of function analysis with…”. This idea should be adequately discussed in the context of the manuscript.

Author Response

Dear Reviewer,

First, I would like to thank you for the positive comments on our manuscript. We have revised the MS according to your suggestions, which improved our MS greatly.

Below is our response to your comments point by point:

Comments and Suggestions for Authors

I had an opportunity to review manuscript “Isolation and Functional Characterization of MsFTa, a FLOWERING LOCUS T homolog from Perennial Forage Alfalfa (Medicago sativa)” for future publication in International Journal Molecular Science. This manuscript by Kang and coauthors describes the key role of FT ortholog in alfalfa in flowering time. Authors performed a functional characterization of MsFTa by expression analysis, subcellular localization studies and generation of over-expression lines.

The project is correctly addressed, results obtained imply interesting conclusions and the manuscript is clear, easy to read and concrete. But minor revisions should be addressed.

-Line 83_94: This paragraph must be included in the legend of figure 1.

R: This paragraph was included in the legend of Figure 1.

-Line 118_123: This paragraph must be included in the legend of figure 2.

R: This paragraph was included in the legend of Figure 2.

-Line 134_136: This paragraph must be included in the legend of figure 3.

R: This paragraph was included in the legend of Figure 3.

-Line 150_153: This paragraph must be included in the legend of figure 4.

R: This paragraph was included in the legend of Figure 4.

-Line 148_150: These lines must be included in the material and methods section.

R: Information of Lines 148-150 was included in the M & M section (Lines 338-389).

-Line 172_173, 192_193, 194_196:  Materials and Methods are described in these lines, which are duplicated in the material and methods section.

R: Lines 172-173, 192-193 and 194-196, were omitted.

-Line 141_144: These sentences must be included in discussion section.

R: Lines 143-144 were removed from Result section.

-Line 162_167: These sentences must be included in discussion section.

R: Lines 162 & 166-167 were removed from Result section.

-Line 215: Mutant fta1-1 should be briefly described in order to make easier the reading of results.

R: Information of mutant fta1-1 was added (Lines 218-219).

-Line 281: “…a loss of function analysis with…”. This idea should be adequately discussed in the context of the manuscript.

R:  Lines 281-283 were rephrased (Lines 286-288).

Reviewer 2 Report

The authors report a study of the florigen function in alfalfa. The novel finding concerns the definition of MsFTa as bona fide orthologue of the Arabidopsis FT gene. Molecular data support this idea and some initial characterisation of MsFTa expression is herewith described. 

The study is well designed, and the paper is well written.  I have no major criticism of the work presented, and in my view the data support the conclusions drawn. 

Author Response

Dear Reviewer,

    Thank you very much for your positive comments on our manuscript.

    Kind regards,

    Zhen Wang